**Data Availability Statement:** All relevant data are available in the paper and its Supporting Information files.

# How curricular changes influence medical students' perceptions of basic science: A pilot study

Yousef Elfanagely[1]*, Joshua Ray Tanzer[2], Ricardo Pulido[3], Hanin Rashid[4], Liesel Copeland[5]

1 Department of Internal Medicine, Brown University, Providence, RI, United States of America, 2 Department of Biostatistics, Brown University, Providence, RI, United States of America, 3 Department of Otolaryngology-Head and Neck Surgery, University of Washington, Seattle, WA, United States of America, 4 Department of Psychiatry, Rutgers Robert Wood Johnson Medical School, New Brunswick, New Jersey, United States of America, 5 Department of Education, Rutgers Robert Wood Johnson Medical School, New Brunswick, New Jersey, United States of America

* yelfanagely@gmail.com

## Abstract

### Theory

The perceived value of study material may have implications on learning and long-term retention. This study compares the perceived value of basic science of medical students from schools with a traditional "2+2" curriculum and the USMLE Step 1 placed before core clerkships to those from medical schools that have undergone curricular revisions, resulting in shortened pre-clerkship curricula and administration of the USMLE Step 1 after core clerkships.

### Hypothesis

We hypothesize that differences in curricula, particularly duration of pre-clerkship curriculum and timing of the USMLE Step 1, affect medical students' perceived value of basic science.

### Methods

A twenty item anonymous questionnaire using a 5-point Likert scale was developed to assess medical students' perceptions of basic science. The questionnaire was distributed to third-year medical students across four medical schools. Generalized linear models and p-values were calculated comparing the perceived value and use of basic science between medical schools with the USMLE Step 1 before clerkships and 2-years of basic science (BC) and medical schools with the USMLE Step 1 after core clerkships and 1.5-years of basic science (AC).

### Results

The questionnaire was distributed to 695 eligible students and completed by 287 students. Students at BC schools tended to view basic science as more essential for clinical practice than students at AC schools across both outcomes (rating independence of basic science

**Funding:** The authors received no specific funding for this work.

**Competing interests:** No authors have competing interests

and clinical practice, AC school mean = 2.97, BC school mean = 2.73, $p$ = 0.0017; rating importance of basic science to clinical practice, AC school mean = 3.30, BC schools mean = 3.50, $p$ = 0.0135).

## Conclusions

Our study suggests that students who have a longer basic science curriculum tend to value basic science greater than students with a shorter basic science curriculum. The timing of the USMLE Step 1 may also influence this relationship. Curricular decisions, such as reductions in pre-clerkship curricula and administration of the USMLE Step 1 after clerkships, may impact medical students' perceptions of the value of basic science to clinical practice. This can have implications on their future engagement with basic science and should be considered when modifying curriculum.

## Introduction

The current structure of medical education can be attributed to the reform following early criticisms of medical schools in Flexner's 1910 report. In his report, Flexner criticized the nonscientific approach and lack of emphasis on basic science in medical courses and research [1–3]. He argued that basic science played a fundamental role in the practice of medicine and should be reflected in medical training [4]. Since Flexner's report, many schools have adopted a "2+2" model with medical students spending their first two years taking basic science courses, such as anatomy and biochemistry, and their last two years completing clinical training in a teaching hospital [5]. In support of Flexner's criticisms, additional studies have provided evidence that basic science is integrated in clinical reasoning [6–10].

While preclinical basic science is a pillar of medical education, the literature assessing the attitudes medical students have towards basic science is incongruent [11, 12]. Some studies cite positive attitudes shared by medical students towards basic science [13–18]. Other studies report students having increasingly negative opinions of basic science during their medical education [19–21]. Assessing the attitudes and subjective value medical students have towards basic science is important because attitude and subjective value of study material influence motivation, deeper learning, and future engagement with material [22–24]. Further investigation is warranted as to what factors may contribute to medical students' perceived value of basic science.

One factor that may contribute to the subjective value of basic science is curriculum design. The two prominent curricular changes that differ from the traditional "2+2" model are the shortening of pre-clerkship courses and the administration of the USMLE Step 1 after completion of core clerkships [25, 26]. The USMLE Step 1 is a standardized exam representing the culmination of applied basic science knowledge acquired during medical school [27]. The USMLE Step 1 is not without its shortcomings. Some medical students adopt a "binge and purge" mentality when studying for the exam as they view its contents as "clinically irrelevant minutiae" [28]. Despite this, the USMLE Step 1 score is the most prominent basic science exam and can be viewed as an indirect measure of short-term retention of basic science material. Medical schools that have changed the timing of the USMLE Step 1 to post-clerkship and reduced the length of pre-clerkship courses have done so to promote more integrated basic science learning in a clinical context [25]. After their curriculum changes, these medical schools have reported higher mean USMLE Step 1 scores [25]. It is unclear, however, if the reported

increase in USMLE Step 1 score is due to curriculum changes as National USMLE scores have steadily increased and the schools involved in the study have had USMLE Step 1 scores historically above average [29].

In this paper, the subjective value of basic science will be assessed in relation to the curriculum design of medical schools. Unlike the USMLE Step 1, which is a short-term goal influenced by career aspirations, the subjective value of basic science can have an important influence on behaviors, including long-term engagement with learning material and retention [24, 30–32]. Long-term retention of basic science is practically important given the frequent licensing exams doctors are required to take, but also empirically important as the value of education depends largely upon its lifespan [33]. Additionally, despite the widespread belief that much of the factual knowledge learned in medical courses is quickly forgotten, this forgetfulness is more accurately described as "temporarily inaccessible," and the information can be recalled when needed in clinical circumstances [33]. Ultimately, we hope to evaluate the relationship between curriculum structure and attitudes towards basic science, and as such contribute to the current debate about curriculum changes.

## Methods

### Materials

We developed a twenty item questionnaire (see Table 1) to assess medical students' perceptions of the value of basic science. The items are presented in a 5-point Likert-scale with scale

**Table 1. Twenty item questionnaire used to assess student's perceived importance of basic science in different settings such as career, classroom, and licensing exams.**

| Item | |
|------|---|
| 1 | A physician can effectively treat most medical problems without knowing the details of the biological processes involved |
| 2 | Most basic science information is so far removed from clinical medicine that its usefulness to the practicing doctor is minimal |
| 3 | One of the most important facets of a good physician is his/her knowledge of biological mechanisms |
| 4 | Applying the basic science of medicine to clinical practice is a skill that should be reinforced early on in medical education |
| 5 | Students do not need to know all the facts of basic science to develop a good working knowledge of basic science |
| 6 | The basic science information I have gained to date is fundamental to my future role as a physician |
| 7 | A deeper understanding in basic science is required to be a good clinical educator |
| 8 | I was overwhelmed with the amount of basic science I was taught |
| 9 | I remember a majority of the basic science material I was taught |
| 10 | I feel inadequate with my knowledge in basic science |
| 11 | I believe a physician with a deep understanding of basic science is a better clinician than a physician with a shallow understanding of basic science |
| 12 | Basic science is the foundation to a good clinical practice |
| 13 | After taking the Step 1 exam, I had a better appreciation for basic science |
| 14 | I try to integrate my basic science knowledge during my clerkships |
| 15 | I believe a physician values basic science content |
| 16 | I believe a physician instinctively integrates basic science in a clinical setting |
| 17 | I believe the best teaching physicians are explicit about basic science found in a clinical setting |
| 18 | I believe basic science only matters for licensing exams (USMLE Step 1) |
| 19 | Clinical knowledge can be acquired without complete understanding of its basic science background |
| 20 | I attempt to identify basic science during my clinical encounters |

**Table 2. Two categories were developed to better encapsulate different attitude about basic sciences.**

| Category | Items in questionnaire |
|---|---|
| Independence of basic science and clinical practice | 1, 2, 12, 5, 19, 20, 8, 10 (Omega total = 0.77) |
| Importance of basic science to clinical practice | 3, 6, 15, 4, 13, 18, 7, 11, 17, 14, 16, 9 (Omega total = 0.89) |

These two categories were drawn from an initial pool of items comparing baseline experiences with basic science, retention of basic science, significance of basic science for career goals, significance of basic science in the classroom, significance of basic science for licensing exams, significance of basic science in becoming a clinical educator, and willingness to integrate or identify basic science concepts during clerkships. The two categories were identified using exploratory factor analysis, represented below with the corresponding questionnaire item(s) in the category. Overall internal consistency for the scale was appropriate (hierarchal omega = 0.76).

points labeled "strongly disagree" (1), "disagree" (2), "neutral" (3), "agree" (4), and "strongly agree" (5). The questionnaire and study protocol were reviewed and approved by the Rutgers University Institutional Review Board (IRB #Pro20160000954).

The items in the questionnaire addressed the role of basic science in medical education (see Table 2). Established techniques for developing questionnaires were used to ensure the items posed in our questionnaire were valid for the purpose of this study [34–36]. Items within the questionnaire were inspired by the surveys constructed by Custers, Alam, and Gupta who also assessed medical students' attitudes towards basic science [13, 19, 20]. After development, the questionnaire was piloted by twenty medical students from two of the four participating medical schools. Students were asked to assess the quality and length of the questionnaire and whether individual questions were repetitive, biased, ambiguous, or confusing. Overall, feedback was positive and only minor corrections were made.

## Distribution

The questionnaire was anonymously distributed to third-year medical students across four medical schools. Names of the institutions were withheld due to the sensitive and exploratory nature of this data. Institution A and B were private medical schools. Institution C and D were public medical schools. The medical schools were chosen because of their different approaches in scheduling the USMLE Step 1 and their relative geographic proximity. All four medical schools were located in the Mid-Atlantic region of the United States (see Table 3). Two medical schools (Institution A and Institution B) have their allotted dedicated study period for the USMLE Step 1 after 12 months of clerkship experience (AC). The two remaining medical schools (Institution C and Institution D) have their allotted dedicated study period for the

**Table 3. Clinical experiences prior to taking USMLE Step 1 of the four medical schools participating in this study.**

| Medical School | Timing of dedicated Step Studying | Months of clinical experience prior to taking USMLE Step 1 | Start of core clerkships (months of basic science) |
|---|---|---|---|
| Institution A | Third year between January and February | 12 | January, Year 2 (16) |
| Institution B | Third year between January and February | 12 | January, Year 2 (16) |
| Institution C | Second year between end of April and late June | 0 | July, Year 3 (24) |
| Institution D | Second year between end of April and late June | 0 | July, Year 3 (24) |

Institution A and Institution B have 12 months of clinical experience scheduled prior to taking the USMLE Step 1. Institution C and Institution D have no clinical experience scheduled prior to taking the USMLE Step 1. For our study, all participants were given the survey after completing at least 6 months of core clerkships.

USMLE Step 1 immediately after 2 years of basic science material (BC). Distribution of the questionnaire occurred from March of third-year to October of fourth-year for medical students at all four medical schools. The timing of questionnaire distribution was chosen because all study participants would have completed their USMLE Step 1 and had at least 6 months of clerkship experience. The link to the on-line questionnaire was distributed via e-mail and promoted via Facebook class pages. Four reminder e-mails were sent after the initial email sharing the link to increase response rate.

## Analysis

The goal of the analysis was to identify whether or not AC schools (Institution A and Institution B) and BC schools (Institution C and Institution D) had meaningful differences in perceptions of the importance of the basic science curriculum. Generalized linear mixed modeling was used because clustered sampling from multiple schools violates the assumption of independence [37]. Random intercepts were estimated by each university to address correlations between errors. The means across the AC schools and BC schools were calculated.

To account for possible confounding, two covariates were added. First, a random effect covariate was added to account for the possibility of differences between AC schools (Institution A and Institution B). Institution B modified their curriculum more recently than Institution A. The random effect was included to allow for differences between these two institutions. Second, due to the ongoing sampling design, participants who completed the questionnaire at different times of the year may have had differing opinions due to the academic calendar. To account for this, a fixed effect of "calendar year" was included.

Before data were analyzed, a priori power analysis was performed based on the equations provided by Fitzmaurice, Laird, and Ware [38]. The sample size used in the analysis included 287 cases. Type one error rate alpha was fixed at the research standard 0.05. Small, medium, and large population effect conditions were assumed by varying the intraclass correlation coefficient (ICC; small = 0.50, medium = 0.30, large = 0.10) and difference between means (small = 0.20, medium = 0.50, and large = 0.80). These are standard recommendations for effect sizes from Cohen (1992). Even when population effects were assumed the hardest to detect (i.e. ICC = 0.50, difference between means = 0.20), power for the analysis was above the research standard of 0.80 (power = 0.90). This suggests that, if there was a true population effect, the analytic design was well equipped to identify it.

## Results

The questionnaire was distributed to a total of 695 students, however only 287 students (41.29%) completed the questionnaire. There were a total of 133 (39.12%) participants from AC schools and 154 (42.25%) participants from BC schools (Table 4). A factor analysis was used to assess the underlying structure of the question items [39]. The first two eigenvalues

**Table 4. Data of the survey participants.**

| | | # of completed survey | Total # of eligible students | Percentage |
|---|---|---|---|---|
| AC | Institution A | 76 | 188 | 40.43% |
| | Institution B | 57 | 152 | 37.50% |
| BC | Institution C | 84 | 166 | 50.60% |
| | Institution D | 70 | 189 | 37.04% |

Total number of participants was 381 from four medical schools. Below is a breakdown of the participants.

**Table 5. Results of data analysis.**

| Category | Items | Avg(AC) | Avg(BC) | Cohen's d | p | Calendar year | p |
|---|---|---|---|---|---|---|---|
| Independence of basic science and clinical practice* | 1, 2, 5, 8, 10, 12, 19, 20 | 2.97 | 2.73 | 0.35 | 0.0017 | <0.01 | 0.2337 |
| Importance of basic science for clinical practice* | 3, 4, 6, 7, 9, 11, 13, 14, 15, 16, 17, 18 | 3.30 | 3.50 | 0.29 | 0.0135 | <0.01 | 0.5359 |

*Statistical significant difference between AC and BC

For data analysis, generalized linear mixed modeling was used with participants nested in universities. Averages for AC and BC schools were compared. Differences between Institution A and Institution B were included as random effect, however differences were minimal (ICC<0.01). Lastly, calendar year was included as confounder. Statistically significant associations are identified.

were greater than 1, suggesting that there may be two clusters of question items. Examining the item loadings with two factors extracted the first set of items (items 1, 2, 5, 8, 10, 12, 19, and 20) which represented the belief that clinical practice may function independently of basic science understanding. The remaining items (items 3, 4, 6, 7, 9, 11, 13, 14, 15, 16, 17, and 18) represented the belief that basic science is important for clinical practice. The description of the categories was based on what each item represented. As a diagnostic check of scale reliability, McDonald's coefficient omega was calculated within the sample using Rstudio psych package [40]. The sample estimate was within the recommended value of 0.80 (hierarchal omega = 0.76) [41]. These two subscales were taken as the outcome variables in the analysis.

Data distributions were examined to ensure that they were appropriately distributed for the analysis. For both subscales, skew and kurtosis was less than |1.0|, suggesting that they were appropriate for the analysis. Models were fit using the Rstudio lme4 package [42]. Residual plots were examined visually and appeared normally distributed; skewness and kurtosis for the residuals was less than |1.0|. Means for the AC and BC school are displayed in Table 5.

Significant differences were identified across both study variables. The rating scale for all items used a 3 as average. Students at AC schools tended to view basic science as less essential for clinical practice than students at BC schools across both outcomes (rating independence of basic science and clinical practice, AC school mean = 2.97, BC school mean = 2.73, $p = 0.0017$; rating importance of basic science to clinical practice, AC school mean = 3.30, BC schools mean = 3.50, $p = 0.0135$). The effect sizes of these differences were moderately small (rating independence of clinical practice, Cohen's d = 0.35; rating importance of basic science to clinical practice, Cohen's d = 0.29).

Next, differences were considered in relation to covariates. Differences between Institution A and Institution B were not large. The ICC for these groups was essentially zero (rating independence of basic science and clinical practice, ICC<0.01; rating importance of basic science to clinical practice, ICC<0.01). This suggests that there was no evidence of differences between Institution A and Institution B despite their differing histories of changing the USMLE Step 1 and clerkship timing. Relationships to calendar year were nonsignificant across both outcomes (rating independence of clinical practice, beta<0.01; $p = 0.2337$; rating importance of basic science to clinical practice, beta<0.01; $p = 0.5359$).

## Discussion

In this pilot study involving four medical schools, we investigated whether curriculum structure has an influence on medical students' perceptions of the value of basic science. Perceptions can have an important influence on behaviors, including acquisition and retention of learning material [30, 31]. Thus, even though self-perception may not always accurately reflect knowledge, this study can inform the current discussions surrounding medical school curricular structures [43].

Medical students who had 2 years of pre-clerkship courses and their USMLE Step 1 exam before clerkships more strongly endorsed the belief that basic science is essential for clinical practice. Results were independent of the timing of transition from the traditional curriculum and when medical students completed the questionnaire. Therefore, it is likely results were reflective of the differences in dedicated time learning basic science and time spent in the hospital.

We hypothesize that schools with a longer basic science curriculum before clerkship rotations may emphasize basic science to a greater extent. Alternatively, curricula with reduced time dedicated to learning basic science and earlier exposure to the hospital in clerkship rotations may reinforce the belief that preparation for clerkship rotations is less reliant on basic science. In Walsh et al, medical students demonstrated significant interest in learning about patients in the context of their disease for their own benefit [44]. More time spent in the hospital may expose students to other factors that aid in clinical decisions and patient-centered care. Students may begin to view clinical experience or evidence based medicine as the crux of clinical practice. Clinical experiences and patient preference can often dictate the management of difficult patient cases that do not follow the textbook features discussed in basic science courses [45].

We hope our findings can contribute to the ongoing discussion of curriculum reform as new challenges, such as changing the USMLE Step 1 exam to a pass/fail grading system, present themselves [46]. Many variables impact medical education, including student perceptions of basic science, which must be considered when adjusting a curriculum [47, 48]. Future studies should further explore the impact of curricular design on students' perceived value. Ultimately, the perceived value of basic science is important because of its implications on long-term retention and processing new material. Moreover, it is critical that students and practitioners are able and willing to transfer their basic science knowledge to clinical practice when encountering novel problem solving [49]. If students do not perceive value in basic science, deeper learning and application may be affected [24].

## Limitations

Our study demonstrates how changes in curricula affect the attitude medical students' have towards basic science. We cannot delineate if the differences observed were secondary to the reduction in pre-clerkship courses or administration of the USMLE Step 1 after core clerkships as both medical schools underwent these curricular changes. Additionally, the study design was observational, which can be limited by confounding variables and cannot determine a cause-and-effect relationship [50]. Data analysis was exploratory and empirically driven. The two categories for our question items were created after a factor analysis. Several steps were taken during the data analysis to limit potential confounding variables. Only 287 students completed the questionnaire out of the 695 eligible students (41.29%). Analysis included measures to ensure nonresponse rate did not significantly influence data. However, low response rates can limit the ability to generalize findings [51]. Data collected also had a response bias in favor of Institution C, with 50.60% of eligible students completing the questionnaire. Demographics, such as age and gender of the participants, were not collected. Additional information as to who completed the questionnaire could have aided in the generalizability of this study. Our data analysis also included measures to ensure there was no statistically significant difference between medical schools with similar curricula, despite adjusting curriculum at different times. However, school culture may have influenced the results gathered. The AC medical schools were private institutions, while the BC medical schools were public institutions. Factors such as incoming undergraduate GPA, tuition, and the quality and volume of

institutional biomedical research can also affect school culture. Furthermore, curricula design in and of itself is part of the culture of a school and changing this may alter the culture. Hence it is important to better understand the influence of curricular design on medical students' values.

## Supporting information

**S1 Data.**
(XLSX)

## Author Contributions

**Conceptualization:** Yousef Elfanagely, Joshua Ray Tanzer, Ricardo Pulido, Hanin Rashid, Liesel Copeland.

**Data curation:** Yousef Elfanagely, Joshua Ray Tanzer, Ricardo Pulido, Hanin Rashid, Liesel Copeland.

**Formal analysis:** Yousef Elfanagely, Joshua Ray Tanzer.

**Investigation:** Yousef Elfanagely, Joshua Ray Tanzer, Hanin Rashid, Liesel Copeland.

**Methodology:** Yousef Elfanagely, Joshua Ray Tanzer, Ricardo Pulido, Hanin Rashid, Liesel Copeland.

**Project administration:** Yousef Elfanagely, Liesel Copeland.

**Resources:** Yousef Elfanagely, Joshua Ray Tanzer, Ricardo Pulido, Liesel Copeland.

**Software:** Joshua Ray Tanzer, Liesel Copeland.

**Supervision:** Hanin Rashid, Liesel Copeland.

**Validation:** Yousef Elfanagely, Hanin Rashid, Liesel Copeland.

**Visualization:** Yousef Elfanagely, Liesel Copeland.

**Writing – original draft:** Yousef Elfanagely, Joshua Ray Tanzer, Hanin Rashid, Liesel Copeland.

**Writing – review & editing:** Yousef Elfanagely, Joshua Ray Tanzer, Hanin Rashid, Liesel Copeland.

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
