## [Decision Letter · Decision Letter 0]

5 May 2020

PONE-D-20-07482

How curricular changes influences medical students’ perceptions of basic science: A pilot study

PLOS ONE

Dear Dr. Elfanagely,

Thank you for submitting your manuscript to PLOS ONE. After careful consideration, we feel that it has merit but does not fully meet PLOS ONE’s publication criteria as it currently stands. Therefore, we invite you to submit a revised version of the manuscript that addresses the points raised during the review process.

We would appreciate receiving your revised manuscript by Jun 19 2020 11:59PM. To enhance the reproducibility of your results, we recommend that if applicable you deposit your laboratory protocols in protocols.io, where a protocol can be assigned its own identifier (DOI) such that it can be cited independently in the future. For instructions see: http://journals.plos.org/plosone/s/submission-guidelines#loc-laboratory-protocols

We look forward to receiving your revised manuscript.

Kind regards,

Frederick Grinnell

Academic Editor

PLOS ONE

"Unfunded study "

5. Please include your tables as part of your main manuscript and remove the individual files. Please note that supplementary tables (should remain/ be uploaded) as separate "supporting information" files

Reviewers' comments:

Reviewer's Responses to Questions

**Comments to the Author**

1. Is the manuscript technically sound, and do the data support the conclusions?

Reviewer #1: Yes

Reviewer #2: Partly

2. Has the statistical analysis been performed appropriately and rigorously? 

Reviewer #1: Yes

Reviewer #2: Yes

3. Have the authors made all data underlying the findings in their manuscript fully available?

Reviewer #1: Yes

Reviewer #2: Yes

4. Is the manuscript presented in an intelligible fashion and written in standard English?

Reviewer #1: Yes

Reviewer #2: Yes

5. Review Comments to the Author

Reviewer #1: PLOS Review

- Key comments:

- This study deserves to be published. The framing is well done, with the authors highlighting the importance of perceptions.

- The authors should explore the extent to which Step 1 encourages learning basic sciences. This ultimately is a question of what the basic sciences are: an approach to problem solving, or a series of factoids?

- The authors MUST address the evidence that AC schools actually see a boost in Step 1 scores (Acad Med. 2019 Mar;94(3):371-377), as this runs counter to their claims.

- There are several grammatical errors. I have done my best to note them.

- Title comment: should be “curricular changes influence” or “curricular change influences.”

- Abstract comments:

- The acronyms in “AC schools” and “BC schools” need clearer introduction. In the methods section, “(USMLE Step 1 before core clerkships and 2 years of pre-clerkship course)” could be edited to “(USMLE Step 1 before core clerkships and 2 years of pre-clerkship course [BC schools]).”

- In results, the total number of eligible students from Table 4 could be stated..

- In the conclusion, “function independent” should be “function independently.”

- In the conclusion, the first two sentences are unclear. If curricular revisions “decrease the belief that clinical practice may function independent of basic science understanding,” then that would suggest students from AC school value basic sciences more, which is the opposite of the authors’ core argument. In the second sentences, is “clinical acumen” meant to be contrasted with appreciation of basic science?

- Introduction comments:

- The authors do a good job of justifying the need for their study, citing the mixed results in the literature (though many of these citations are not from US medical schools, and so their relevance is questionable).

- “As a prominently basic science exam” could be changed to “As the most prominent basic science exam…”

- “indirect measure retention of basic science material” should be “indirect measure of retention of basic science material”

- The relationship of Step 1 to basic sciences is controversial. Its content has been described as “clinically irrelevant minutiae” requiring a “binge and purge mentality” (JAMA. 2014;311(13):1358-1359). In my opinion, basic science is about a way of thinking and problem solving, more so than about memorization of factoids that are promptly forgotten. In this respect, the authors already do a good job by qualifying Step 1 as “an indirect measure,” and I agree with them that it is the best tool we have for this purpose. In my opinion, the paper would benefit from a sentence or two about the limitations of using Step 1 as a reflection of basic science.

- “despite the widespread belief much of the factual knowledge” should be “despite the widespread belief that much of the factual knowledge.”

- Regarding the claim that “there will be less objective measures to assess retention and perceived value for the material taught in medical school”: Step 1 is a snapshot of knowledge, and as such does not measure retention. Of note, here is a paper that seems to address retention of knowledge (Adv Physiol Educ. 2014 Dec;38(4):315-20). I also think it is a stretch to say that Step 1 score is a measure of perceived value of basic sciences; I believe the main thing it reflects is the career ambitions of the test taker.

- I would replace “If the study provides evidence for differing perceptions of the relevance of basic science, the results can contribute to the current debate about curriculum changes” with something like “The present study seeks to evaluate the relationship of curriculum structure with attitudes toward basic sciences, and as such contributes to the current debate about curriculum changes.”

- Methods comments:

- “The first two eigenvalues were greater and 1” should be “The first two eigenvalues were greater than 1”?

- I am unable to comment on factor analysis as I do not have experience with this method.

- To me the two beliefs (independence of basic science and clinical practice, and importance of basic science and clinical practice) are tautology. Furthermore, the authors do not state how they linked the items in the questionnaire to these categories. For example, item 13 (“After taking the Step 1 exam, I had a better appreciation for basic science”) does not clearly suggest the “importance of basic science to clinical practice.” The paper would benefit from further definition and distinction of these terms, and further explanation on how questionnaire items were linked to the two main categories.

- The distribution section is very clear. It would also be interesting to compare the survey results to student scores on Step 1 (ostensibly representing mastery of basic sciences), but the work this would take may be prohibitive.

- “Random intercepts were estimated by university do address correlations between errors” is unclear.

- Analysis appears rigorous to me, though I do not have experience with biostats.

- Results comments:

- “This reduced the sample size to 287 student” should be “This reduced the sample size to 287 students”

- Discussion comments:

- The authors assume that perception translates into retention of material. However, a study they cite (Acad Med. 2019 Mar;94(3):371-377) suggests that AC schools have higher Step 1 scores and lower failure rates. Assuming that Step 1 is the gold standard for basic science assessment, this data runs counter to the authors’ claims. The authors MUST address this.

- The authors could further reflect on the relatively small effect size.

- Limitations comments:

- Data collected also had a response bias in favor of Institution D may, with 41.49% eligible students completing the survey.

- Good exploration of limitations.

- Tables comments:

- Table 2: “Independence of clinical practice” should be something like “Independence of basic science and clinical practice.”

Reviewer #2: The manuscript describes an initial look at student perception of the independence and value of basic science to clinical practice as a function of differences in the length of the pre-clerkship and whether Step 1 is administered before or after core clerkships. As the current COVID-19 pandemic is illustrating, the importance of basic science in clinical practice is continuing to grow, yet it is not clear that student perception of this value is growing with it. Measures of student perception of basic science are thus valuable for curriculum design/reform.

Major concerns/critiques:

1) The groups have two key differences: the AC group has a 3-semester pre-clerkship and administers Step 1 following core clerkships, while the BC group has a 4-semester pre-clerkship and administers Step 1 prior to core clerkships. It is not clear whether a shortened pre-clerkship, a later Step 1 or both influence student perception of the value/independence of basic science to clinical practice. Better would have been four groups, separating each of the two variables.

2) It is not clear whether covariants were fully accounted for. The AC group was comprised of two private institutions. The BC group was comprised of two public institutions. Additional possible covariants include tuition cost (higher cost may encourage interest in specialties), mean student MCAT scores at matriculation (higher MCATs indicate better initial basic science knowledge and possibly basic science interest), and quality/volume of institutional biomedical research (more and better opportunities for student research).

Minor concerns/critiques:

1) A comparison of Step 1 scores would be of value and should be referenced to Step 2. While Step 1 was administered at different points in the curricula, Step 2 was likely administered at a similar point. If Step 2 scores are similar, differences in Step 1 become more meaningful. A better Step 1 score may be indicative of a higher perceived value of basic science knowledge.

2) The statement in the conclusion of the abstract "Curricular revisions, such as reduction in preclerkship curricula and administration of the USMLE Step 1 after clerkships, may decrease the belief that clinical practice may function independent of basic science understanding." may be a misstatement. The data of the article correlates a shorter preclerkship/later Step 1 with worse student perception of the value of basic science for clinical practice. Better might be "..., may decrease the belief that effective clinical practice depends upon basic science understanding."

6. PLOS authors have the option to publish the peer review history of their article (what does this mean?). If published, this will include your full peer review and any attached files.

Reviewer #1: Yes: David Roy Chen

Reviewer #2: No

---

## [Author Response · Author response to Decision Letter 0]

27 Jun 2020

In response to the reviewers’ comments, we addressed each point. 

Major concerns:

1. The groups have two key differences: the AC group has a 3-semester pre-clerkship and administers Step 1 following core clerkships, while the BC group has a 4-semester pre-clerkship and administers Step 1 prior to core clerkships. It is not clear whether a shortened pre-clerkship, a later Step 1 or both influence student perception of the value/independence of basic science to clinical practice. Better would have been four groups, separating each of the two variables.

Response: The two trends in medical curricula changes are reduced pre-clerkship experience and administration of Step 1 after core clerkships. Unfortunately, these changes are often done simultaneously. The medical schools that participated in the study had undergone both changes, reducing pre-clerkship courses and administering Step 1 after core clerkships. We were unable to assess whether reducing pre-clerkship courses or administering Step 1 after core clerkships was responsible for our findings. However, our study demonstrates that changes in curricula affect the attitude a medical student has towards basic science. We have added this to the limitations section of our paper. 

2. It is not clear whether covariants were fully accounted for. The AC group was comprised of two private institutions. The BC group was comprised of two public institutions. Additional possible covariants include tuition cost (higher cost may encourage interest in specialties), mean student MCAT scores at matriculation (higher MCATs indicate better initial basic science knowledge and possibly basic science interest), and quality/volume of institutional biomedical research (more and better opportunities for student research).

Response: We attempted to address covariants during the data analysis by including calendar year and timing of curriculum change. We acknowledge that there are additional covariants. As a result, we have added tuition cost, mean student MCAT score, and volume of research to the limitations section of our paper when discussing the significance of school culture. It is difficult to predict how MCAT score or tuition would have affected a medical student’s value of basic science. For example, MCAT score may not have influence the study findings as our measurements occurred after students participated in medical school and similar match rates were found in all four schools.

Minor concerns: 

3. A comparison of Step 1 scores would be of value and should be referenced to Step 2. While Step 1 was administered at different points in the curricula, Step 2 was likely administered at a similar point. If Step 2 scores are similar, differences in Step 1 become more meaningful. A better Step 1 score may be indicative of a higher perceived value of basic science knowledge.

Response: The authors agree that a comparison of Step 1 scores with respect to Step 2 CK scores would add value to the discussion. However, the study collected anonymous data from students and did not ask for Step1 or Step 2 CK scores. When the initial questionnaire was drafted, we were concerns about asking for Step 1 and Step 2 CK scores. We felt that if students were asked this type of sensitive data they could be less inclined to complete the survey and there would be no way of confirming their scores. In addition, we felt that the perceived long-term retention of basic science may differ regardless of Step 1 score. 

4. The statement in the conclusion of the abstract "Curricular revisions, such as reduction in preclerkship curricula and administration of the USMLE Step 1 after clerkships, may decrease the belief that clinical practice may function independent of basic science understanding." may be a misstatement. The data of the article correlates a shorter preclerkship/later Step 1 with worse student perception of the value of basic science for clinical practice. Better might be "..., may decrease the belief that effective clinical practice depends upon basic science understanding."

Response: The conclusion was entirely revised to address these concerns. 

Concerns about the title: 

5. should be “curricular changes influence” or “curricular change influences.”

Response: The title has been changed to “How curricular changes influence medical students’ perceptions of basic science: A pilot study”

Concerns about the abstract:

6. The acronyms in “AC schools” and “BC schools” need clearer introduction. In the methods section, “(USMLE Step 1 before core clerkships and 2 years of pre-clerkship course)” could be edited to “(USMLE Step 1 before core clerkships and 2 years of pre-clerkship course [BC schools]).”

Response: The acronyms “AC schools” and “BC schools” were more clearly introduced in the revised methods section of the abstract. The sentence now reads “Generalized linear models and p-values were calculated comparing the perceived value and use of basic science between medical schools with the USMLE Step 1 before clerkships and 2-years of basic science (BC) and medical schools with the USMLE Step 1 after core-clerkships and 1.5-years of basic science (AC).”

7. In results, the total number of eligible students from Table 4 could be stated. 

Response: The total number of eligible students is now included in the first sentence of the methods section of the abstract. 

8. In the conclusion, the first two sentences are unclear. If curricular revisions “decrease the belief that clinical practice may function independent of basic science understanding,” then that would suggest students from AC school value basic sciences more, which is the opposite of the authors’ core argument. In the second sentences, is “clinical acumen” meant to be contrasted with appreciation of basic science?

Response: The conclusion was entirely revised to address these concerns. 

9. “Function independent” should be “function independently.”

Response: The conclusion was entirely revised to address concerns about the first two sentences being unclear. The phrase “function independent” is no longer in the abstract’s conclusion. 

10. The authors do a good job of justifying the need for their study, citing the mixed results in the literature (though many of these citations are not from US medical schools, and so their relevance is questionable).

Response: The number of citations in the introduction was expanded and included additional American references to address this concern and strengthen the relevance of this study. 

11. “As a prominently basic science exam” could be changed to “As the most prominent basic science exam…”

Response: The sentence has been changed to reflect the grammatical concern. 

Concerns about the introduction: 

12. "indirect measure retention of basic science material” should be “indirect measure of retention of basic science material”

Response: The sentence has been changed to reflect the grammatical concern.

13. The relationship of Step 1 to basic sciences is controversial. Its content has been described as “clinically irrelevant minutiae” requiring a “binge and purge mentality” (JAMA. 2014;311(13):1358-1359). In my opinion, basic science is about a way of thinking and problem solving, more so than about memorization of factoids that are promptly forgotten. In this respect, the authors already do a good job by qualifying Step 1 as “an indirect measure,” and I agree with them that it is the best tool we have for this purpose. In my opinion, the paper would benefit from a sentence or two about the limitations of using Step 1 as a reflection of basic science.

Response: Two sentences were added in the introduction to acknowledge the opinions of some medical students towards the USMLE Step 1 exam. 

14. "despite the widespread belief much of the factual knowledge” should be “despite the widespread belief that much of the factual knowledge.”

Response: The sentence has been changed to reflect the grammatical concern.

15. Regarding the claim that “there will be less objective measures to assess retention and perceived value for the material taught in medical school”: Step 1 is a snapshot of knowledge, and as such does not measure retention. Of note, here is a paper that seems to address retention of knowledge (Adv Physiol Educ. 2014 Dec;38(4):315-20). I also think it is a stretch to say that Step 1 score is a measure of perceived value of basic sciences; I believe the main thing it reflects is the career ambitions of the test taker.

Response: This sentence was ultimately removed to ensure a clear message at the end of the introduction. 

16. I would replace “If the study provides evidence for differing perceptions of the relevance of basic science, the results can contribute to the current debate about curriculum changes” with something like “The present study seeks to evaluate the relationship of curriculum structure with attitudes toward basic sciences, and as such contributes to the current debate about curriculum changes.”

Response: The last sentence of the introduction was modified to reflect the advice provided by the reviewer. 

Concerns with the methods section: 

17. “The first two eigenvalues were greater and 1” should be “The first two eigenvalues were greater than 1”?

Response: The sentence has been changed to reflect the grammatical concern. 

18.To me the two beliefs (independence of basic science and clinical practice, and importance of basic science and clinical practice) are tautology. Furthermore, the authors do not state how they linked the items in the questionnaire to these categories. For example, item 13 (“After taking the Step 1 exam, I had a better appreciation for basic science”) does not clearly suggest the “importance of basic science to clinical practice.” The paper would benefit from further definition and distinction of these terms, and further explanation on how questionnaire items were linked to the two main categories.

Response: We acknowledge the reviewers point about the similarity between the two beliefs. While it seems intuitive that the two beliefs are highly related, and this was demonstrated within the data, we did not know this until completion of the analysis. This was an exploratory and empirically driven analysis that included a variety of questions related to beliefs about science. The definition of the categories was based on which questions tended to correlate with each other in the factor analysis, and the delineation of belief scales were decided upon by the researchers based on what the items appeared to represent. To address this in the text, we added further description of how the factor analysis was interpreted. We also acknowledged this point as a limitation. 

19. “Random intercepts were estimated by university do address correlations between errors” is unclear.

Response: Random intercepts were estimated by university to address correlations between errors, which would be a violation of regression assumptions. While the goal of this analysis was not to compare each university individually, it was expected that for whatever unknown reasons each university would likely have differences in means. A random effect allows the model estimation algorithm to adjust for these expected differences without directly comparing the means.

Concerns with data analysis: 

20. Experiments must have been conducted rigorously, with appropriate controls, replication, and sample sizes.

Response: We have updated all tables to reflect the true number of participants who were included in the study. As stated in the initial manuscript, two sites were removed because of low response rates (response rates less than 30%). To make the information clearer to the reader, we removed this from the revised manuscript. We have provided our data as well. 

Concerns with the results section:

21. “This reduced the sample size to 287 student” should be “This reduced the sample size to 287 students”

Response: The sentence has been changed to reflect the grammatical concern.

Concerns with the discussion section:

22. The authors assume that perception translates into retention of material. However, a study they cite (Acad Med. 2019 Mar;94(3):371-377) suggests that AC schools have higher Step 1 scores and lower failure rates. Assuming that Step 1 is the gold standard for basic science assessment, this data runs counter to the authors’ claims. The authors MUST address this.

Response: The authors agree that currently Step 1 is typically used as the gold standard for assessing basic science material. The study we cited by Jurich et.al (Acad Med. 2019 Mar;94(3):371-377)) does indeed show that the AC schools had higher Step 1 scores and lower failure rates. However, one of the major limitations from the cited study is that the initial Step 1 scores from those schools involved in the study were historically above average to other schools’ Step 1 scores nationally. As such it is challenging to compare the Step 1 scores of the AC schools and BC schools as a true indicator of retention of basic science material. It is also worth noting that the medical students who attended the medical schools with shortened preclerkship curricula had higher incoming MCAT scores and GPAs (AC’s MCAT 35.6 and GPA 3.81 vs. BC’s MCAT 32 and GPA 3.68). Several studies have reflected already that these differences themselves can have some correlation to how students perform on standardized tests such as Step 1. Our hope with this study was to assess whether there was any notable differences in the perceived value of basic science based on changes in curriculum. The implications of this difference may have an effect on long-term retention that far extends beyond Step 1. Step 1 is a measure of basic science after an intensive study period. Though it is a strong measure of basic science knowledge, students who do not value basic science may fail to use their basic science consistently in their careers unless they are in a specialty where it is very evident. 

23. The authors could further reflect on the relatively small effect size.

Response: Our study suggests that differences in curriculum may impact medical students’ perceived relevance of basic science. It is important to remember that all students had a level of value of basic science, yet interesting to note that those in curricular with more time studying basic science prior to significant clinical experience had a relatively higher value. This demonstrates some of the ways that differing styles of instruction may relate to differing trends in opinions.

Concerns with the tables:

24. “Independence of clinical practice” should be something like “Independence of basic science and clinical practice.”

Response: Table 2 was revised to reflect this feedback.

---

## [Editor Report · Decision Letter 1]

7 Jul 2020

How curricular changes influence medical students’ perceptions of basic science: A pilot study

PONE-D-20-07482R1

Dear Dr. Elfanagely,

We’re pleased to inform you that your manuscript has been judged scientifically suitable for publication and will be formally accepted for publication once it meets all outstanding technical requirements.

Kind regards,

Frederick Grinnell

Academic Editor

PLOS ONE
---

## [Editor Report · Acceptance letter]

9 Jul 2020

PONE-D-20-07482R1 

How curricular changes influence medical students’ perceptions of basic science: A pilot study 

Dear Dr. Elfanagely:

I'm pleased to inform you that your manuscript has been deemed suitable for publication in PLOS ONE. Congratulations! Your manuscript is now with our production department. 

Kind regards, 

on behalf of

Professor Frederick Grinnell 

Academic Editor

PLOS ONE